# Characterizing the Lipid Profile in Patients with Vertebral or Hip Fragility Fractures: A Hospital-Based Descriptive Study

**DOI:** 10.3390/jcm14197029

**Published:** 2025-10-04

**Authors:** Yan Feng, Qinghua Tang, Siyu Li, Lei Yang, Ming Yang, Jiancheng Yang, Yuhong Zeng

**Affiliations:** Department of Osteoporosis, Honghui Hospital, Xi’an Jiaotong University, Xi’an 710054, China; fengyan19772@163.com (Y.F.); gshua691994@163.com (Q.T.); 13698001246@163.com (S.L.); yleii82@126.com (L.Y.); 18591776708@163.com (M.Y.)

**Keywords:** fragility fractures, osteoporosis, total cholesterol, triglycerides, high-density lipoprotein cholesterol, low-density lipoprotein cholesterol, vertebral fractures, hip fractures

## Abstract

**Background/Objectives:** Fragility fractures, particularly in the vertebra and hip, are a significant health concern in the elderly, often associated with osteoporosis. Emerging evidence suggests a link between lipid profiles and bone health, but the characteristics of lipid biomarkers in patients with fragility fractures remain underexplored. **Methods:** This study analyzed serum lipid biomarkers, including total cholesterol (TC), triglycerides (TG), high-density lipoprotein cholesterol (HDL-C), and low-density lipoprotein cholesterol (LDL-C) in 10,540 patients aged 50 and older with either vertebral or hip fragility fractures. We compared lipid levels between the two fracture groups and examined the relationship between lipid profiles and baseline characteristics of patients. **Results:** Patients with vertebral fractures exhibited significantly higher serum levels of TC, TG, HDL-C, and LDL-C compared to those with hip fractures. These differences remained statistically significant after adjusting for confounding variables. Multivariable regression analysis revealed that age was inversely associated with TC, TG, and LDL-C, but positively associated with HDL-C. All lipid levels were significantly higher in women than in men. Time from fracture to admission and BMI were positively associated with TG levels and inversely associated with HDL-C. Vertebral fracture patients had a higher prevalence of abnormally high TC (3.03% vs. 0.78%), TG (9.15% vs. 3.54%), and LDL-C (2.80% vs. 1.04%), but lower prevalence of abnormally low HDL-C (20.53% vs. 26.66%; *p* < 0.001 for all). **Conclusions:** Our findings highlight distinct lipid profile characteristics in patients with vertebral and hip fragility fractures, suggesting that physiological or metabolic changes following different fracture types may differentially influence lipid metabolism. These insights may inform targeted prevention and management strategies for fragility fractures.

## 1. Introduction

Fragility fractures, also known as fragility fractures, represent the most severe consequence of osteoporosis. It was estimated that the global population aged 50 years and older at risk of fragility fractures was 158 million in 2010, and this number is projected to double by 2040 [1]. In 2013, the International Osteoporosis Foundation (IOF) reported that a fragility fracture occurs globally every three seconds [2]. Fragility fractures have become a major global public health concern. They not only cause pain and disability, significantly reducing patients’ quality of life, but fractures at specific sites, particularly the vertebra and hip, also increase mortality risk and impose substantial medical and economic burdens on healthcare systems [3,4].

Numerous clinical risk factors associated with fracture risk have been identified, including low bone mineral density (BMD), prior fracture, sex (female), early menopause, advanced age, low body weight, and glucocorticoid treatment [5]. Furthermore, serum lipid profiles have been suggested as potential factors linked to osteoporosis and fragility fractures. Previous meta-analyses indicated that the use of lipid-lowering agents was associated with increased BMD at various sites and a reduced risk of fragility fractures, supporting a connection between serum lipids and osteoporosis [6,7,8]. In recent years, there has been growing insight into the relationships between blood lipid levels, metabolism, and osteoporosis [9,10,11]. Associations between BMD, metabolic syndrome, and cardiovascular events have been reported, and lipid profiles may play a significant role in these links [12,13,14,15].

Many observational studies have investigated the relationship between specific lipid biomarkers and BMD or fracture risk [16,17,18,19,20,21,22]. These biomarkers include total cholesterol (TC), triglycerides (TG), high-density lipoprotein cholesterol (HDL-C), and low-density lipoprotein cholesterol (LDL-C). However, to the best of our knowledge, the characteristics of the serum lipid profiles in patients who have already sustained a fragility fracture remain unreported. The vertebra and hip are common sites for fragility fractures. In this study, we focused on patients aged 50 years and older with vertebral or hip fragility fractures, aiming to analyze the characteristics of serum cholesterol levels (TC, TG, HDL-C, and LDL-C) in this fracture population. We further examined the correlations between these lipid biomarkers and age, sex, body mass index (BMI), time from fracture to hospital, and fracture sites.

## 2. Materials and Methods

### 2.1. Study Population

This is a single-center retrospective study. Patients with vertebral or hip fragility fractures were identified from the Fragility Fracture Database established by Honghui Hospital, Xi’an Jiaotong University [23,24]. We exported information (epidemiological data, medical history, and lipid biochemical indicators) of patients hospitalized from January 2020 to August 2023 from the database. Patients were included in this study based on the following criteria: (1) Age ≥ 50 years and postmenopausal status (for females); (2) Fracture sustained due to low energy trauma (Falls from standing height or lower, and daily activities such as bending over, coughing, and sneezing) or no identifiable trauma; (3) Confirmed diagnosis of vertebral or hip fragility fracture; (4) Availability of lipid biomarker measurements. Exclusion criteria were: (1) Fractures at sites other than the vertebra or hip; (2) Duration exceeding 3 months between fracture occurrence and hospital admission; (3) Pathological fractures secondary to malignancy. The study protocol received approval from the Institutional Review Board of Honghui Hospital, Xi’an Jiaotong University (Approval No. 202305003), and informed consent was waived due to the retrospective nature of the study utilizing deidentified data.

### 2.2. Measurement of Lipid Biomarkers and Definition of Abnormalities

Fasting blood samples were collected in the morning within 24 h of admission and centrifuged to obtain serum. Serum levels of TC, TG, HDL-C, and LDL-C were measured preoperatively for all enrolled fracture patients using a Roche Cobas 6000 biochemical analyzer (Roche, Basel, Switzerland), with all assays performed using original manufacturer’s reagents and following standard calibration and quality control procedures. Abnormal lipid biomarkers were defined according to the reference ranges recommended by the Chinese guideline for lipid management [25]:TC: Optimal level < 5.2 mmol/L; High level ≥ 6.2 mmol/LTG: Optimal level < 1.7 mmol/L; High level ≥ 2.3 mmol/LHDL-C: Low level < 1.0 mmol/LLDL-C: Optimal level < 3.4 mmol/L; High level ≥ 4.1 mmol/L

Lipid biomarkers were considered abnormal if they fell into the high or low categories defined above.

### 2.3. Statistical Analysis

Categorical data are presented as numbers and percentages (%) and were compared between groups using the Chi-square (*χ*^2^) test. Normally distributed continuous data are expressed as mean ± standard deviation (SD) and were compared using Student’s *t* test. Continuous variables not conforming to a normal distribution are presented as median (interquartile range, IQR), and group comparisons were performed using the Mann–Whitney U test. Multivariate linear regression analysis was used to determine the association of lipid markers with age, sex, BMI, fracture sites, time from fracture to hospital, smoking history, drinking history, history of fractures, history of cancer, diabetes, anemia, and coronary disease. Multivariate logistic regression analysis was used to compare the risk of abnormal blood lipid between patients with vertebral and hip fractures. All statistical analyses were performed using IBM SPSS Statistics version 27.0 (IBM Corp., Armonk, NY, USA). To mitigate potential false positives resulting from the large sample size, a *p*-value < 0.01 was defined as statistically significant.

## 3. Results

### 3.1. Characteristics of the Study Population

As shown in Figure 1, based on the inclusion and exclusion criteria, this study ultimately included 10,540 patients with vertebral and hip fractures.

The mean age of the 5817 patients with vertebral fragility fractures was 68.61 years, while the mean age of the 4723 patients with hip fragility fractures was 75.78 years (Table 1). This difference in age between the two groups was statistically significant (*p* < 0.001). Compared to hip fracture patients, the proportion of males was significantly lower among vertebral fracture patients (29.43% vs. 35.36%, *p* < 0.001). Vertebral fracture patients had higher BMI and a longer time from fracture to hospital. The proportions of patients with a history of fractures and cancer were significantly higher in the vertebral fracture group. Conversely, hip fracture patients had a higher prevalence of diabetes, anemia, and coronary disease. There were no significant differences between the two fracture groups regarding the proportions of patients with a history of smoking and drinking.

### 3.2. The Differences in Lipid Biomarkers Between Patients with Vertebral Fracture and Hip Fracture

Student’s *t* test showed patients with vertebral fracture had higher serum levels of TC, TG, HDL-C, and LDL-C compared to hip fracture patients (Table 2). These differences were also validated in a multiple linear regression model with or without adjusting for variables, including age, sex, BMI, time from fracture to hospital, history of fractures, history of cancer, diabetes, anemia, and coronary disease.

### 3.3. Blood Lipid Characteristics of Patients with Vertebral and Hip Fragility Fractures

To evaluate the characteristics of lipid profiles in patients with vertebral and hip fragility fractures, we used multiple linear regression to examine the associations of TC, TG, HDL-C, and LDL-C with variables including age, sex, BMI, time from fracture to hospital, smoking history, drinking history, history of fractures, history of cancer, diabetes, anemia, and coronary disease (Table 3).

The levels of TC, TG, and LDL-C displayed a negative association with age in patients with vertebral fracture or hip fracture (*p* < 0.001) in Model 1 (no covariates were adjusted). After adjusting for confounders (Model 2), a negative association was still present and statistically significant. However, HDL-C levels were positively associated with age in Model 1 and Model 2.

Using females as the reference, regression analyses—both unadjusted and adjusted—consistently showed that all lipid marker levels were significantly lower in males than in females (*p* < 0.001).

Regarding BMI, although Model 1 indicated significant positive associations between serum TC, LDL-C and BMI, these associations lost significance after adjustment for covariates. In contrast, TG showed a significant positive association with BMI, while HDL-C was negatively associated, both with and without adjustment.

Similarly to BMI, time from fracture to hospital was positively associated with TC and LDL-C in unadjusted models, but these associations ceased to be significant after adjustment. However, a consistently positive association was observed between time from fracture to hospital and TG regardless of adjustment. Interestingly, while no significant association was found between time from fracture to hospital and HDL-C in unadjusted models (*p* = 0.207), a significant negative relationship emerged after adjustment (*p* = 0.002).

Compared to patients with a history of smoking, those without exhibited only lower TC and HDL-C levels in unadjusted analyses; these differences became non-significant after adjustment. No significant associations were observed between smoking history and serum TG or LDL-C levels. Furthermore, all lipid indicators showed no significant associations with drinking history, prior fracture, or cancer—either with or without adjustment.

Relative to non-diabetic patients, those with diabetes demonstrated lower TC, HDL-C, and LDL-C levels, but higher TG levels. Compared to patients without anemia or coronary disease, those with either condition exhibited lower serum TC and LDL-C levels both with and without adjustment, while lower TG and HDL-C were observed only in unadjusted models.

### 3.4. Differences in Lipid Biomarkers Among Patients with Vertebral and Hip Fractures in Different Populations

To minimize the influence of other variables on the differences in lipid profiles between patients with vertebral and hip fractures, we conducted a stratified analysis based on various factors, as presented in Table 4. The results demonstrated that serum TC levels were consistently significantly higher in vertebral fracture patients than in hip fracture patients across all subgroups, regardless of whether other variables were adjusted or not. Regarding TG levels in vertebral fracture patients, after adjusting for covariates, significant differences compared to hip fracture patients were observed only in subgroups with BMI > 24.9 kg/m^2^ and in those with diabetes; in all other subgroups, TG levels remained significantly higher. For serum HDL-C levels, no significant differences were found between vertebral and hip fracture patients in subgroups including age < 80 years, females, BMI < 18.5 kg/m^2^, time from fracture to admission ≤ 72 h, and those with coronary disease. Significant differences in LDL-C levels between the two groups were absent only among patients with coronary disease; in all other subgroups, vertebral fracture patients exhibited significantly higher LDL-C levels than hip fracture patients.

### 3.5. Comparison of Proportions with Abnormal Lipid Indicators Between Vertebral and Hip Fracture Patients

As shown in Table 5, the proportion of patients with abnormally high levels was significantly greater in the vertebral fracture group compared to the hip fragility fracture group for TC (3.03% vs. 0.78%), TG (9.15% vs. 3.54%), and LDL-C (2.80% vs. 1.04%). Conversely, the proportion of patients with abnormally low HDL-C levels was significantly lower among vertebral fracture patients than hip fracture patients (20.53% vs. 26.66%).

To eliminate the potential influence of other variables on the proportion of abnormal lipid indicators, we converted lipid values into dichotomous variables (normal vs. abnormal) and performed multivariate logistic regression analysis (Table 6). Using vertebral fracture as the reference group, the risk of abnormally high levels of TC (OR: 0.260, *p* < 0.001), TG (OR: 0.609, *p* < 0.001), and LDL-C (OR: 0.458, *p* < 0.001) remained significantly lower in hip fracture patients after adjusting for confounding variables, including age, sex, BMI, time from fracture to hospital, smoking history, drinking history, history of fractures, history of cancer, diabetes, anemia, and coronary disease. In contrast, hip fracture patients exhibited a significantly higher risk of abnormally low HDL-C levels compared to those with vertebral fractures (OR: 1.459, *p* < 0.001).

## 4. Discussion

This study provided a novel insight into the serum lipid profiles of elderly patients with vertebral and hip fragility fractures, revealing significant differences in lipid biomarker levels and their associations with age, sex, BMI, time from fracture to hospital, and fracture site. These findings contribute to the growing body of evidence linking lipid metabolism to osteoporosis and fracture risk, with implications for both clinical practice and future research.

The elevated lipid levels in vertebral fracture patients compared to hip fracture patients may partly be explained by age differences, as vertebral fracture patients were significantly younger (mean age 68.61 vs. 75.78 years). Previous studies have shown that TC, TG, and LDL-C levels tend to decrease with advancing age, while HDL-C increases, consistent with our correlation findings [26,27]. However, these differences persisted even after adjusting for age and other covariates, suggesting that additional factors—such as fracture-specific pathophysiological mechanisms or risk profiles—may also play a role.

The age-related patterns of lipid biomarkers align with established literature. The negative association of TC, TG, and LDL-C with age, and the positive correlation of HDL-C with age, mirror trends observed in the general population [28]. In the context of fragility fractures, this implies that older patients, who are at greater risk of fractures, generally have lower TC, TG, and LDL-C levels but higher HDL-C. However, the role of HDL-C in bone health remains complex. Hussain et al. [16] reported that higher HDL-C levels were associated with increased fracture risk in older adults, which may relate to our finding of higher HDL-C in younger vertebral fracture patients. This paradox suggests that HDL-C’s impact on fracture risk may depend on fracture type, age, or comorbidities, warranting further exploration.

Sex differences in lipid profiles were also notable, with females exhibiting higher levels of all lipid biomarkers compared to males. This is consistent with post-menopausal lipid changes, where women often show elevated HDL-C and sometimes TC levels [29]. As our study included only postmenopausal women, these findings align with expected hormonal influences on lipid metabolism. Similarly, the positive association between TG and BMI reflects the well-documented association between higher body mass and dyslipidemia [30]. Conversely, serum levels of HDL-C showed a significant negative correlation with BMI, indicating that patients with higher body weight tend to have lower HDL-C levels. Indeed, well-established studies have demonstrated that obesity adversely affects HDL-C metabolism, resulting in reduced serum HDL-C concentrations [31,32,33].

Based on our findings, the observed differences in lipid profiles between vertebral and hip fracture patients may be importantly influenced by acute-phase responses following fracture. The significantly shorter time from fracture to admission in the hip fracture group, coupled with the positive association between longer time and higher TC, TG, and LDL-C levels—as well as the negative association with HDL-C—suggests that systemic inflammation in the early post-fracture period may transiently suppress lipid levels. Indeed, studies have found that among hospitalized older adults, patients with acute-phase symptoms exhibit lower levels of TC, LDL-C, and HDL-C compared to those without such symptoms [34]. Moreover, in both subgroups with time from fracture to hospital ≤72 h and >72 h, patients with vertebral fractures consistently exhibited higher levels of TC, TG, and LDL-C compared to those with hip fractures, suggesting that the differences in lipid profiles between the two fracture types may not be entirely attributable to acute-phase changes, but rather reflect underlying metabolic differences between these patient populations. The delayed emergence of a significant difference in HDL-C only in the >72 h subgroup further implies that HDL-C may be more sensitive to acute inflammatory suppression, and that differential regulation or recovery becomes apparent only after the initial phase. Although our study attempted to account for timing-related confounding through stratification and regression adjustment, the absence of direct inflammatory markers represents a limitation. Future studies incorporating biomarkers such as CRP or IL-6 could help clarify the interplay between fracture type, inflammation, and lipid metabolism.

The higher prevalence of abnormally high TC, TG, and LDL-C in vertebral fracture patients, coupled with a lower prevalence of abnormally low HDL-C, underscores the distinct lipid profiles between fracture groups. These differences may have implications for cardiovascular risk management, given dyslipidemia’s role in cardiovascular disease [35]. Mechanistically, lipids may influence bone health through inflammation, oxidative stress, or effects on osteoblast and osteoclast activity [36,37,38]. For example, high LDL-C and low HDL-C are linked to increased inflammation, which may exacerbate bone loss [39,40]. Our findings thus contribute to the growing evidence of a lipid-bone health nexus, as supported by prior studies linking lipid profiles to osteoporosis and fracture risk [41,42,43].

Clinically, these results suggest that lipid management strategies might need tailoring based on fracture site. The higher lipid levels in vertebral fracture patients, who are younger and may have a longer life expectancy, could justify more aggressive lipid-lowering interventions. In contrast, hip fracture patients, who are older and often frailer, may require a more conservative approach. Future studies should explore whether modulating lipid levels can influence fracture risk or recovery outcomes.

This study has several limitations. First, its retrospective design limits causal inferences regarding lipid biomarkers and fracture risk. Prospective studies are needed to elucidate whether lipid abnormalities precede or result from fractures. Second, the lack of BMD measurements in our cohort precludes direct assessment of the relationship between lipid profiles and BMD. Future studies should incorporate dual-energy X-ray absorptiometry (DXA) to clarify these associations. Third, our analysis did not account for lipid-lowering medication use, which could influence lipid levels and fracture risk. Detailed medication histories should be included in future research. Finally, the study population was limited to Chinese patients, and findings may not generalize to other ethnic groups with different lipid metabolism profiles.

## 5. Conclusions

This study systematically characterized the lipid profiles of older adults with vertebral and hip fragility fractures. We revealed that patients with vertebral fractures had higher levels of TC, TG, HDL-C, and LDL-C compared to those with hip fractures. These differences remained significant even after adjusting for other variables, suggesting that physiological and metabolic changes following fractures at different sites may exert distinct influences on lipid metabolism. Further research is essential to clarify these relationships and assess the potential of lipid-modifying interventions in fracture prevention and management.

## Figures and Tables

**Figure 1 jcm-14-07029-f001:**
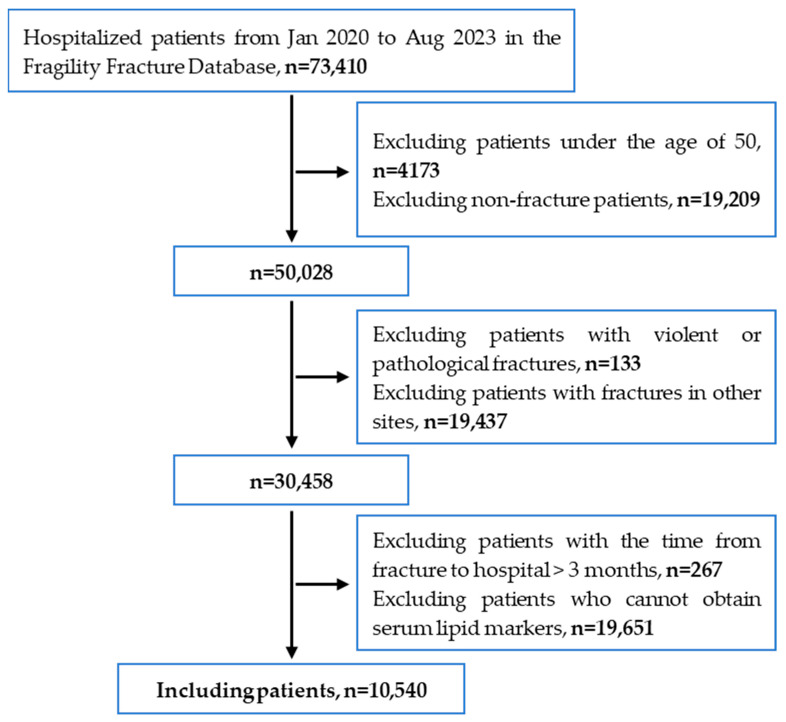
Patient screening flowchart for this study.

**Table 1 jcm-14-07029-t001:** Characteristics of the study population.

		Total	Vertebral Fracture	Hip Fracture	*p*
Total, *n*		10,540	5817	4723	N/A
Age (years), mean ± SD		71.82 ± 11.51	68.61 ± 10.71	75.78 ± 11.23	<0.001
Men, *n* (%)		3382	1712 (29.43)	1670 (35.36)	<0.001
Women	*n* (%)	7158	4105 (70.57)	3053 (64.64)	<0.001
	Menopausal age, mean ± SD	49.68 ± 2.32	49.65 ± 2.34	49.95 ± 1.17	0.347
BMI (kg/m^2^), mean ± SD		22.05 ± 3.84	22.49 ± 3.84	21.54 ± 3.77	<0.001
Time from fracture to hospital (d), IQR		3 (1, 10)	6 (2, 15)	1 (0.375, 3)	<0.001
Smoking history, *n* (%)		650 (6.17)	379 (6.52)	271 (5.74)	0.099
Drinking history, *n* (%)		266 (2.52)	147 (2.53)	119 (2.52)	0.981
History of fractures, *n* (%)		804 (7.63)	523 (8.99)	281 (5.95)	<0.001
History of cancer, *n* (%)		403 (3.82)	262 (4.50)	141 (2.99)	<0.001
Diabetes, *n* (%)		1670 (15.84)	702 (12.07)	968 (20.50)	<0.001
Anemia, *n* (%)		312 (2.96)	46 (0.79)	266 (5.63)	<0.001
Coronary disease, *n* (%)		2175 (20.64)	780 (13.41)	1395 (29.54)	<0.001

**Table 2 jcm-14-07029-t002:** The differences in lipid biomarkers between patients with vertebral fracture and hip fracture.

	Total	Vertebral Fracture	Hip Fracture	Model 1 (Ref. Hip Fracture)			Model 2 (Ref. Hip Fracture)		
				*β*	*95% CI*	*p*	*β*	*95% CI*	*p*
TC (mmol/L), mean ± SD	4.04 ± 0.97	4.27 ± 0.96 *	3.74 ± 0.90	0.280	0.229, 0.331	<0.001	0.198	0.145, 0.251	<0.001
TG (mmol/L), mean ± SD	1.29 ± 0.68	1.40 ± 0.73 *	1.15 ± 0.56	0.199	0.162, 0.236	<0.001	0.130	0.091, 0.169	<0.001
HDL-C (mmol/L), mean ± SD	1.23 ± 0.33	1.25 ± 0.32 *	1.21 ± 0.33	0.056	0.038, 0.074	<0.001	0.056	0.036, 0.076	<0.001
LDL-C (mmol/L), mean ± SD	2.33 ± 0.81	2.50 ± 0.80 *	2.12 ± 0.75	0.220	0.177, 0.263	<0.001	0.140	0.095, 0.185	<0.001

* *p* < 0.01 vs. Hip fracture (Student’s *t* test). Model 1: no covariates were adjusted. Model 2: age, sex (male; female), BMI, time from fracture to hospital, history of fractures (yes; no), history of cancer (yes; no), diabetes (yes; no), anemia (yes; no), and coronary disease (yes; no) were adjusted.

**Table 3 jcm-14-07029-t003:** Multivariate linear regression analysis of lipid profile characteristics in patients with fragility fractures.

	TC				TG				HDL-C				LDL-C			
	Model 1		Model 2		Model 1		Model 2		Model 1		Model 2		Model 1		Model 2	
	*β (95% CI)*	*p*	*β (95% CI)*	*p*	*β (95% CI)*	*p*	*β (95% CI)*	*p*	*β (95% CI)*	*p*	*β (95% CI)*	*p*	*β (95% CI)*	*p*	*β (95% CI)*	*p*
Age	−0.155 (−0.157, −0.153)	<0.001	−0.081 (−0.083, −0.079)	<0.001	−0.182 (−0.183, −0.180)	<0.001	−0.156 (−0.158, −0.154)	<0.001	0.072 (0.071, 0.073)	<0.001	0.090 (0.089, 0.091)	<0.001	−0.172 (−0.173, −0.170)	<0.001	−0.106 (−0.108, −0.104)	<0.001
Sex (Ref. female)	−0.268 (−0.307, −0.229)	<0.001	−0.266 (−0.317, −0.215)	<0.001	−0.119 (−0.148, −0.090)	<0.001	−0.117 (−0.154, −0.080)	<0.001	−0.224 (−0.238, −0.210)	<0.001	−0.232 (−0.250, −0.214)	<0.001	−0.199 (−0.232, −0.166)	<0.001	−0.196 (−0.241, −0.151)	<0.001
BMI	0.039 (0.033, 0.045)	0.003	0.008 (0.002, 0.013)	0.545	0.087 (0.083, 0.091)	<0.001	0.052 (0.048, 0.056)	<0.001	−0.043 (−0.045, −0.041)	0.001	−0.042 (−0.044, −0.040)	0.002	0.042 (0.036, 0.048)	0.002	0.014 (0.008, 0.020)	0.274
Time from fracture to hospital	0.119 (0.117, 0.121)	<0.001	0.025 (0.023, 0.027)	0.059	0.130 (0.128, 0.132)	<0.001	0.081 (0.080, 0.082)	<0.001	−0.013 (−0.014, −0.012)	0.207	−0.042 (−0.042, −0.041)	0.002	0.101 (0.099, 0.103)	<0.001	0.018 (0.016, 0.020)	0.181
Smoking history	−0.067 (−0.177, 0.043)	<0.001	0.005 (−0.113, 0.123)	0.723	−0.008 (−0.086, 0.070)	0.573	0.013 (−0.075, 0.101)	0.409	−0.085 (−0.122, 0.048)	<0.001	−0.007 (−0.050, 0.036)	0.651	−0.034 (−0.126, 0.058)	0.012	0.014 (−0.088, 0.116)	0.357
Drinking history	−0.037 (−0.194, 0.120)	0.006	0.011 (−0.152, 0.174)	0.448	0.000 (−0.112, 0.112)	0.971	0.006 (−0.116, 0.128)	0.702	−0.045 (−0.098, 0.008)	0.001	0.023 (−0.036, 0.082)	0.126	−0.021 (−0.152, 0.110)	0.135	0.003 (−0.138, 0.144)	0.834
History of fractures (Ref. no)	0.018 (−0.053, 0.089)	0.067	−0.004 (−0.082, 0.090)	0.720	0.032 (0.017, 0.081)	0.001	0.004 (−0.060, 0.069)	0.731	−0.013 (−0.037, 0.011)	0.201	−0.004 (−0.035, 0.027)	0.785	−0.012 (−0.071, 0.045)	0.229	−0.010 (−0.084, 0.064)	0.442
History of cancer (Ref. no)	0.016 (−0.084, 0.116)	0.107	0.031 (−0.095, 0.156)	0.012	0.014 (−0.057, 0.085)	0.151	0.020 (−0.072, 0.112)	0.134	−0.015 (−0.048, 0.018)	0.130	−0.007 (−0.052, 0.038)	0.603	0.014 (−0.071, 0.098)	0.156	0.027 (−0.053, 0.163)	0.039
Diabetes (Ref. no)	−0.073 (−0.124, −0.022)	0.001	−0.040 (−0.106, 0.027)	0.001	0.120 (0.083, 0.157)	<0.001	0.146 (0.097, 0.195)	<0.001	−0.105 (−0.123, −0.087)	<0.001	−0.111 (−0.135, −0.087)	<0.001	−0.081 (−0.124, −0.038)	<0.001	−0.043 (−0.102, 0.016)	0.001
Anemia (Ref. no)	−0.118 (−0.232, 0.004)	<0.001	−0.067 (−0.212, 0.078)	<0.001	−0.063 (−0.141, 0.015)	<0.001	−0.020 (−0.126, 0.086)	0.122	−0.038 (−0.078, 0.001)	<0.001	−0.020 (−0.126, 0.086)	0.122	−0.111 (−0.207, −0.015)	<0.001	−0.061 (−0.184, 0.062)	<0.001
Coronary disease (Ref. no)	−0.159 (−0.206, −0.112)	<0.001	−0.104 (−0.167, −0.039)	<0.001	−0.059 (−0.092, 0.026)	<0.001	−0.014 (−0.063, 0.035)	0.288	−0.030 (−0.046, 0.014)	0.003	−0.022 (−0.046, 0.002)	0.116	−0.162 (−0.201, −0.103)	<0.001	−0.111 (−0.168, −0.054)	<0.001

Model 1: no covariates were adjusted. Model 2: age, sex (male; female), BMI, fracture sites (vertebra; hip), time from fracture to hospital, smoking history (yes; no), drinking history (yes; no), history of fractures (yes; no), history of cancer (yes; no), diabetes (yes; no), anemia (yes; no), and coronary disease (yes; no) were adjusted.

**Table 4 jcm-14-07029-t004:** Stratified analysis of differences in lipid profiles between patients with vertebral fractures and hip fractures.

		Model 1 (Ref. Hip Fracture)			Model 2 (Ref. Hip Fracture)		
		*β*	*95% CI*	*p*	*β*	*95% CI*	*p*
TC							
Age (years)	≤65	0.235	0.145, 0.325	<0.001	0.169	0.079, 0.260	<0.001
	66–79	0.251	0.167, 0.335	<0.001	0.169	0.081, 0.258	<0.001
	≥80	0.278	0.182, 0.374	<0.001	0.244	0.148, 0.340	<0.001
Sex	Female	0.259	0.196, 0.322	<0.001	0.197	0.134, 0.260	<0.001
	Male	0.286	0.206, 0.366	<0.001	0.228	0.148, 0.308	<0.001
BMI (kg/m^2^)	<18.5	0.216	0.098, 0.334	<0.001	0.155	0.037, 0.273	<0.001
	18.5–24.9	0.301	0.236, 0.366	<0.001	0.225	0.160, 0.290	<0.001
	>24.9	0.261	0.151, 0.371	<0.001	0.160	0.050, 0.270	<0.001
Time from fracture to hospital	≤72 h	0.259	0.188, 0.330	<0.001	0.195	0.124, 0.266	<0.001
	>72 h	0.248	0.154, 0.342	<0.001	0.172	0.078, 0.266	<0.001
Diabetes	Yes	0.206	0.071, 0.341	<0.001	0.175	0.040, 0.310	<0.001
	no	0.285	0.230, 0.340	<0.001	0.201	0.146, 0.256	<0.001
Coronary disease	Yes	0.129	0.043, 0.215	<0.001	0.069	0.017, 0.155	0.006
	no	0.294	0.239, 0.349	<0.001	0.226	0.171, 0.281	<0.001
TG							
Age (years)	≤65	0.118	0.045, 0.191	<0.001	0.089	0.016, 0.162	<0.001
	66–79	0.201	0.140, 0.262	<0.001	0.150	0.085, 0.215	<0.001
	≥80	0.174	0.115, 0.233	<0.001	0.153	0.094, 0.212	<0.001
Sex	Female	0.212	0.165, 0.259	<0.001	0.155	0.108, 0.202	<0.001
	Male	0.140	0.081, 0.199	<0.001	0.084	0.025, 0.143	<0.001
BMI (kg/m^2^)	<18.5	0.229	0.150, 0.307	<0.001	0.156	0.078, 0.234	<0.001
	18.5–24.9	0.210	0.165, 0.255	<0.001	0.162	0.117, 0.206	<0.001
	>24.9	0.131	0.039, 0.223	<0.001	0.044	−0.048, 0.136	0.149
Time from fracture to hospital	≤72 h	0.154	0.107, 0.201	<0.001	0.073	0.026, 0.120	<0.001
	>72 h	0.108	0.035, 0.181	<0.001	0.057	−0.016, 0.130	0.005
Diabetes	Yes	0.129	0.015, 0.243	<0.001	0.057	−0.057, 0.171	0.130
	no	0.233	0.196, 0.270	<0.001	0.146	0.109, 0.183	<0.001
Coronary disease	Yes	0.198	0.120, 0.276	<0.001	0.117	0.039, 0.195	<0.001
	no	0.194	0.153, 0.235	<0.001	0.130	0.089, 0.171	<0.001
HDL-C							
Age (years)	≤65	0.071	0.040, 0.102	0.003	0.031	0.000, 0.062	0.199
	66–79	0.057	0.028, 0.086	0.011	0.035	0.004, 0.066	0.135
	≥80	0.098	0.063, 0.133	<0.001	0.088	0.053, 0.123	<0.001
Sex	Female	0.021	−0.003, 0.045	0.215	0.029	0.005, 0.053	0.112
	Male	0.075	0.048, 0.102	0.002	0.115	0.088, 0.142	<0.001
BMI (kg/m^2^)	<18.5	−0.037	−0.080, 0.006	0.263	−0.020	−0.063, 0.023	0.572
	18.5–24.9	0.083	0.059, 0.107	0.002	0.076	0.052, 0.100	<0.001
	>24.9	0.080	0.041, 0.119	0.006	0.068	0.029, 0.107	0.008
Time from fracture to hospital	≤72 h	0.028	0.003, 0.053	0.135	0.048	0.023, 0.073	0.022
	>72 h	0.252	0.221, 0.283	<0.001	0.229	0.198, 0.220	<0.001
Diabetes	Yes	0.074	0.031, 0.117	0.003	0.120	0.077, 0.163	<0.001
	no	0.039	0.029, 0.059	0.008	0.045	0.025, 0.065	0.005
Coronary disease	Yes	−0.003	−0.048, 0.042	0.848	−0.004	−0.049, 0.041	0.825
	no	0.067	0.045, 0.089	<0.001	0.068	0.046, 0.090	<0.001
LDL-C							
Age (years)	≤65	0.185	0.109, 0.261	<0.001	0.132	0.056, 0.208	<0.001
	66–79	0.183	0.110, 0.256	<0.001	0.108	0.032, 0.184	<0.001
	≥80	0.202	0.122, 0.282	<0.001	0.172	0.092, 0.252	<0.001
Sex	Female	0.199	0.146, 0.252	<0.001	0.137	0.084, 0.190	<0.001
	Male	0.229	0.160, 0.298	<0.001	0.158	0.089, 0.227	<0.001
BMI (kg/m^2^)	<18.5	0.160	0.060, 0.260	<0.001	0.095	0.005, 0.195	0.006
	18.5–24.9	0.233	0.178, 0.288	<0.001	0.155	0.100, 0.210	<0.001
	>24.9	0.221	0.127, 0.315	<0.001	0.133	0.039, 0.227	<0.001
Time from fracture to hospital	≤72 h	0.219	0.158, 0.280	<0.001	0.161	0.100, 0.222	<0.001
	>72 h	0.162	0.082, 0.242	<0.001	0.090	0.010, 0.170	<0.001
Diabetes	Yes	0.157	0.040, 0.275	<0.001	0.124	0.006, 0.242	<0.001
	no	0.223	0.176, 0.270	<0.001	0.141	0.094, 0.188	<0.001
Coronary disease	Yes	0.083	−0.027, 0.193	0.011	0.037	−0.073, 0.147	0.308
	no	0.229	0.182, 0.276	<0.001	0.163	0.116, 0.210	<0.001

Model 1: no covariates were adjusted. Model 2: age, sex (male; female), BMI, time from fracture to hospital, smoking history (yes; no), drinking history (yes; no), history of fractures (yes; no), history of cancer (yes; no), diabetes (yes; no), anemia (yes; no), and coronary disease (yes; no) were adjusted.

**Table 5 jcm-14-07029-t005:** Percentage and comparison of abnormal lipid biomarkers in patients with fragility fractures.

Biomarkers	Total	Vertebral Fracture	Hip Fracture	*χ* ^2^	*p*
TC, n (%)	213 (2.02)	176 (3.03)	37 (0.78)	57.96	<0.001
TG, n (%)	699 (6.63)	532 (9.15)	167 (3.54)	115.90	<0.001
HDL-C, n (%)	2453 (23.27)	1194 (20.53)	1259 (26.66)	64.74	<0.001
LDL-C, n (%)	212 (2.01)	163 (2.80)	49 (1.04)	33.00	<0.001

**Table 6 jcm-14-07029-t006:** Multiple Logistic Regression Results.

	Odds Ratio	*95% CI*	*p*
TC	0.260	0.154, 0.439	<0.001
TG	0.609	0.470, 0.788	<0.001
HDL-C	1.459	1.258, 1.692	<0.001
LDL-C	0.458	0.291, 0.720	<0.001

## Data Availability

The datasets generated and analyzed during the current study are not publicly available due to the limitations of our hospital’s information security and confidentiality system, but data are available from the corresponding author (yjchhyy2021@163.com or xahhzyh@163.com) with the appropriate approval from the Institutional Review Board of Honghui Hospital, Xi’an Jiaotong University.

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
