# Peer review of "Characterizing the Lipid Profile in Patients with Vertebral or Hip Fragility Fractures: A Hospital-Based Descriptive Study"

_jcm, 2025, doi:10.3390/jcm14197029_

Round 1
Reviewer 1 Report
Comments and Suggestions for Authors
General comment:
This is a large and homogeneous cohort, and the topic is interesting and relevant. The study is generally well conducted, but it requires substantial statistical re-evaluation with the support of an experienced statistician to produce more robust results and conclusions—particularly through appropriate multivariate analyses and adjustment for baseline variables.
Specific comments:
- Title, Abstract, Introduction – Appropriate. No major changes required.
- Materials and Methods
- Specify the study design at the beginning (e.g., retrospective, single-center).
- Move the number of patients included to the Results section.
- Clearly describe data collection methodology, stating whether information was obtained retrospectively from medical records or using another approach.
- Statistical Analysis – This is the major weakness of the manuscript.
- Univariate analyses and baseline adjustments between groups (e.g., age, sex, BMI, history of diabetes) are missing.
- Adjustment for baseline variables is essential in observational studies to control for confounding, increase precision, and ensure group comparability in the absence of randomization.
- Continuous variables (e.g., age, BMI, biochemical values) should be included as covariates, and categorical variables as fixed factors in multivariate models.
- Results –
- Split Table 1 into two: (1) baseline variables; (2) lipid parameters (TC, LDL, HDL, TG).
- This second table should report both crude differences between groups (with p-values) and adjusted differences (with 95% CI) after controlling for baseline variables.
- Interpret accordingly: if a difference is significant only before adjustment, it is likely explained by baseline differences; if it remains significant after adjustment, it may reflect a true difference in lipid profiles between groups.
- Report observed power, as non-significant findings in large samples may still hold clinical relevance.
- Move Table 2 to the Supplementary Material. Clarify correlation analyses—Pearson’s r indicates bivariate correlations, which should be presented pairwise without adjustment. If covariate adjustment was applied, this constitutes multivariate analysis or regression, which should be reported with coefficients rather than r.
- Group comparisons (e.g., sex, smoking, alcohol use, diabetes, cardiovascular disease) should use appropriate between-group tests (e.g., Student’s t test), not bivariate correlations.
- In the Statistical Analysis section, specify how correlation strength was classified (e.g., Altman’s thresholds: negligible, weak, moderate, substantial, strong). For multivariate models, indicate effect size (small, medium, large).
- In Table 4, replace the “Outlier” column (already defined in Methods) with a “Total” column reporting the overall percentage of outliers.
- Discussion – Reconsider and potentially revise statements such as “The higher prevalence of abnormally high TC, TG, and LDL in spinal fracture patients, coupled with a lower prevalence of abnormally low HDL, underscores the distinct lipid profiles between fracture groups” after adjusting for baseline variables, as these differences may be attenuated or lost following proper statistical control.
Author Response
This is a large and homogeneous cohort, and the topic is interesting and relevant. The study is generally well conducted, but it requires substantial statistical re-evaluation with the support of an experienced statistician to produce more robust results and conclusions—particularly through appropriate multivariate analyses and adjustment for baseline variables.
Response: We appreciate your positive feedback on our study. In response, we have revised the statistical approach by incorporating multivariable analyses, including multiple linear regression and multiple logistic regression, with adjustments for baseline variables.
Specify the study design at the beginning (e.g., retrospective, single-center).
Response: We have specified our study is a single-center retrospective study at the beginning.
Move the number of patients included to the Results section.
Response: We have moved the number of patients included to the Results section.
Clearly describe data collection methodology, stating whether information was obtained retrospectively from medical records or using another approach.
Response: Thank you for your suggestion. The data in this study were retrospectively obtained from our established fragility fracture database, which allows for the export of all patient medical record information in the form of an Excel document. In this study, we exported information such as epidemiological data, medical history, and lipid biochemical indicators of the patients. This has been described in the Results section. Moreover, we have added a patient screening flowchart, as shown in Figure 1.
Statistical Analysis – This is the major weakness of the manuscript.
Univariate analyses and baseline adjustments between groups (e.g., age, sex, BMI, history of diabetes) are missing.
Adjustment for baseline variables is essential in observational studies to control for confounding, increase precision, and ensure group comparability in the absence of randomization.
Continuous variables (e.g., age, BMI, biochemical values) should be included as covariates, and categorical variables as fixed factors in multivariate models.
Response: Thank you for your suggestion. We have re-analyzed the data using a multiple linear regression model, controlling for other variables.
Split Table 1 into two: (1) baseline variables; (2) lipid parameters (TC, LDL, HDL, TG).
This second table should report both crude differences between groups (with p-values) and adjusted differences (with 95% CI) after controlling for baseline variables.
Interpret accordingly: if a difference is significant only before adjustment, it is likely explained by baseline differences; if it remains significant after adjustment, it may reflect a true difference in lipid profiles between groups
Response: Thank you for your suggestion. We have split Table 1 into two separate tables. In Table 2, we have reported both the p-values and 95% CIs for the analyses with and without adjusting for variables.
Report observed power, as non-significant findings in large samples may still hold clinical relevance.
Response: Thank you for your suggestion. We acknowledge the importance of reporting observed power, particularly in the context of non-significant results from large samples. However, due to the lack of established standards or commonly referenced criteria for interpreting the magnitude of observed power in our specific field, we have chosen not to include it in this revision to avoid potential misinterpretation.
Move Table 2 to the Supplementary Material. Clarify correlation analyses—Pearson’s r indicates bivariate correlations, which should be presented pairwise without adjustment. If covariate adjustment was applied, this constitutes multivariate analysis or regression, which should be reported with coefficients rather than r.
Response: Thank you for your suggestion. We have removed the Pearson correlation analysis and replaced it with a multivariable regression analysis, with the β coefficients reported accordingly.
Group comparisons (e.g., sex, smoking, alcohol use, diabetes, cardiovascular disease) should use appropriate between-group tests (e.g., Student’s t test), not bivariate correlations.
Response: Thank you for your suggestion. In the baseline characteristics presented in Table 1, we have applied Student’s t-test to assess differences in continuous variables between groups.
In the Statistical Analysis section, specify how correlation strength was classified (e.g., Altman’s thresholds: negligible, weak, moderate, substantial, strong). For multivariate models, indicate effect size (small, medium, large).
Response: Thank you for this helpful suggestion. We agree on the importance of clarifying the interpretation of correlation strength and effect sizes in multivariate models. However, due to the absence of universally accepted benchmarks or field-specific guidelines for classifying these measures in our research context, we have opted to report the quantitative values (e.g., regression estimates) without further categorical classification to avoid subjective interpretation. We are open to incorporating specific standards should you be able to provide relevant references
In Table 4, replace the “Outlier” column (already defined in Methods) with a “Total” column reporting the overall percentage of outliers.
Response: Thank you for your suggestion. We have replaced the “Outlier” column with a “Total” column.
Discussion – Reconsider and potentially revise statements such as “The higher prevalence of abnormally high TC, TG, and LDL in spinal fracture patients, coupled with a lower prevalence of abnormally low HDL, underscores the distinct lipid profiles between fracture groups” after adjusting for baseline variables, as these differences may be attenuated or lost following proper statistical control.
Response: Thank you for your suggestion. We employed a multivariable logistic regression model, controlling for other variables, and found that the difference in dyslipidemia proportions between patients with spinal and hip fractures did not diminish or disappear after adjustment. Therefore, we have retained our original statements.
Reviewer 2 Report
Comments and Suggestions for Authors
This is a large single-center study comparing lipid profiles between vertebral and hip fragility fracture patients. The topic is clinically relevant and the cohort size is a clear strength. However, several issues in data consistency, methods transparency, and statistical analysis limit the reliability of the current conclusions.
Major comments
-
Data integrity & table consistency (top priority)
-
Recheck all counts and summaries. For example, anemia totals in Table 1 do not match subgroup sums; please correct and audit all tables.
-
Standardize summary formats (e.g., consistently report median [IQR] or mean ± SD); ensure units and denominators are explicit.
-
Variables meaningful only for subsets (e.g., menopausal age) should be restricted to that subset with the corresponding N clearly indicated.
-
Ensure each table’s headers/columns include all variables for both groups; current column misalignments/omissions hinder interpretation.
-
Remove duplicated/overlapping results sections (age correlations appear twice); consolidate to a single, coherent section.
-
Methods clarity & reproducibility
-
Specify the enrollment period (start/end dates), inclusion/exclusion criteria, and the definition of low-energy trauma used.
-
Provide clear definitions (and data sources) for comorbidities (e.g., ICD codes vs. clinician diagnosis vs. lab thresholds).
-
Detail the timing and conditions of blood draws (fasting status; time from fracture to sampling) and how these were handled analytically.
-
Report the analyzers/assays precisely (model, calibration/quality control in brief).
-
Align and justify abnormal lipid thresholds in text and tables (e.g., HDL-C typically low side only; TC/LDL/TG high side).
-
Statistical analysis: move beyond unadjusted/partial correlations
-
For your primary question (between-site differences), run multivariable linear regression for each lipid (dependent variable) with fracture site as the key predictor, adjusting for age, sex, BMI, time from fracture to sampling, major comorbidities (prior fracture, cancer, diabetes, CAD, anemia), smoking, alcohol, and lipid-lowering medication use. Report adjusted mean differences (β) with 95% CIs and standardized effects.
-
For “abnormal lipid” outcomes, use multivariable logistic regression and report adjusted odds ratios (aOR, 95% CI).
-
Address distributional issues (e.g., TG often skewed): consider log-transformations or robust methods as appropriate.
-
Control for multiple testing (e.g., Benjamini–Hochberg FDR) and prespecify primary vs. exploratory analyses.
-
Add sensitivity analyses: (a) exclude statin users or adjust for dose/class; (b) stratify/adjust by time from fracture to sampling (e.g., ≤72 h vs. >72 h); (c) sex-specific analyses.
-
Confounding & limitations
-
Lipid-lowering therapy and acute-phase changes after trauma are major potential confounders; integrate them explicitly in the analyses and discussion.
-
Bone mineral density (BMD) is absent but clinically important—acknowledge this limitation early, and, if feasible, include proxies (prior fracture history, FRAX items).
-
In Abstract/Conclusions, avoid causal wording; emphasize associations and the potential influence of post-fracture physiological changes on lipid values.
Overall, the manuscript is readable, but a light–moderate language edit would improve clarity and consistency once the structural/table issues are resolved. Key points:
-
Tense and voice: Use past tense for Methods/Results (“We analyzed…”, “HDL-C was…”) and present tense for established knowledge in Introduction/Discussion. Keep active voice where possible.
-
Terminology consistency: Standardize to HDL-C/LDL-C throughout; use one term for the population (e.g., “fragility fractures” or “osteoporotic fractures”) and one for sites (“vertebral” vs “hip”) consistently.
-
Abbreviations: Define at first mention (e.g., BMD, CAD) and use consistently. Provide an abbreviation list.
-
Statistical phrasing: Reserve “significant” for results meeting the alpha threshold; otherwise write “higher/lower but not statistically significant (p=…)”. Prefer “associated with” over “related with.”
-
Grammar & style: Check subject–verb agreement, article use (“the vertebral fracture group was…”), and prepositions (“correlation between X and Y”). Avoid unnecessary slashes (use “and/or” sparingly).
-
Numbers & units: Follow SI/ journal style: space between number and unit (e.g., “mmol/L”), en-dashes for ranges (e.g., 65–80), uniform decimal places, and consistent % formatting.
-
Hyphenation/line breaks: Remove PDF-induced split words (e.g., “hos-pital”), unify hyphen vs en-dash usage, and avoid mid-line hyphen breaks.
-
Tables/Figures: Make captions self-explanatory and parallel in structure; ensure column headers and units are explicit and consistent.
Author Response
This is a large single-center study comparing lipid profiles between vertebral and hip fragility fracture patients. The topic is clinically relevant and the cohort size is a clear strength. However, several issues in data consistency, methods transparency, and statistical analysis limit the reliability of the current conclusions.
Response: Thank you for your positive feedback on our work. We have accordingly revised the statistical methods in response to your comments.
Data integrity & table consistency (top priority)
Recheck all counts and summaries. For example, anemia totals in Table 1 do not match subgroup sums; please correct and audit all tables.
Response: Thank you for pointing out this error. We have corrected it and have also thoroughly reviewed the other data to ensure accuracy.
Standardize summary formats (e.g., consistently report median [IQR] or mean ± SD); ensure units and denominators are explicit.
Response: Thank you for your suggestion. We have thoroughly reviewed all data throughout the manuscript to ensure internal consistency and have verified that all units and denominators are clearly indicated.
Variables meaningful only for subsets (e.g., menopausal age) should be restricted to that subset with the corresponding N clearly indicated.
Response: Thank you for your suggestion. We have categorized “menopausal age” under the “Women” subset, as shown in Table 1.
Ensure each table’s headers/columns include all variables for both groups; current column misalignments/omissions hinder interpretation.
Response: Thank you for your suggestion. We have corrected all table’s headers/columns.
Remove duplicated/overlapping results sections (age correlations appear twice); consolidate to a single, coherent section.
Response: Thank you for your suggestion. We have corrected the results.
Methods clarity & reproducibility
Specify the enrollment period (start/end dates), inclusion/exclusion criteria, and the definition of low-energy trauma used.
Response: Thank you for your suggestion. We have added the start and end dates of enrollment in the methodology section and provided a clear definition of low-energy fracture.
Provide clear definitions (and data sources) for comorbidities (e.g., ICD codes vs. clinician diagnosis vs. lab thresholds).
Response: Thank you for your suggestion. We have specified the source of the data (our established fragility fracture database) in the Methods section.
Detail the timing and conditions of blood draws (fasting status; time from fracture to sampling) and how these were handled analytically.
Response: Thank you for your suggestion. We have added details regarding the timing and conditions of blood sampling in the Methods section.
Report the analyzers/assays precisely (model, calibration/quality control in brief).
Response: Thank you for your suggestion. As requested, we have now included the precise analytical information in the Methods section.
Align and justify abnormal lipid thresholds in text and tables (e.g., HDL-C typically low side only; TC/LDL/TG high side).
Response: Thank you for this important comment. We have carefully reviewed and aligned the descriptions of abnormal lipid thresholds throughout the manuscript, particularly in the Methods section. All cut-off values (e.g., lower threshold for HDL-C, upper thresholds for TC, LDL-C, and TG) are now explicitly defined and consistently applied in the Methods section, based on widely recognized clinical guidelines.
Statistical analysis: move beyond unadjusted/partial correlations
For your primary question (between-site differences), run multivariable linear regression for each lipid (dependent variable) with fracture site as the key predictor, adjusting for age, sex, BMI, time from fracture to sampling, major comorbidities (prior fracture, cancer, diabetes, CAD, anemia), smoking, alcohol, and lipid-lowering medication use. Report adjusted mean differences (β) with 95% CIs and standardized effects.
Response: Thank you for your suggestion. We have removed the partial correlation analysis and replaced it with a multiple linear regression model, as presented in Table 2 and Table 3.
For “abnormal lipid” outcomes, use multivariable logistic regression and report adjusted odds ratios (aOR, 95% CI).
Response: Thank you for your suggestion. We have employed multivariable logistic regression to analyze the abnormal lipid data and have reported the adjusted OR values along with their 95% CI, as shown in Table 6.
Address distributional issues (e.g., TG often skewed): consider log-transformations or robust methods as appropriate.
Response: Thank you for your valuable suggestion regarding the distributional characteristics of triglyceride (TG) levels. We have carefully evaluated the potential skewness in the TG data. However, after performing comparative analyses, we found that the results obtained from conventional multivariate regression were highly consistent with those from log-transformed or robust regression approaches in terms of effect direction and statistical inference. Therefore, to maintain consistency with our analytical framework and facilitate interpretation of the results, we have retained the original method in the present analysis.
Control for multiple testing (e.g., Benjamini–Hochberg FDR) and prespecify primary vs. exploratory analyses.
Response: Thank you for raising this important point regarding multiple testing correction and the distinction between primary and exploratory analyses. We agree that controlling the false discovery rate can be valuable in studies involving numerous hypotheses. In our study, the analyses were primarily hypothesis-driven and focused on a predefined set of variables based on clinical relevance. We found that the overall interpretation of our results remained consistent without FDR adjustment. Furthermore, the sample size and the exploratory nature of certain secondary analyses may make rigorous multiple-testing corrections overly conservative, potentially masking meaningful clinical associations. We have therefore chosen to present unadjusted p-values for transparency, but have interpreted the findings with caution, emphasizing effect sizes and confidence intervals in addition to statistical significance. We are open to applying a multiple testing correction method if the editor and reviewers consider it essential.
Add sensitivity analyses: (a) exclude statin users or adjust for dose/class; (b) stratify/adjust by time from fracture to sampling (e.g., ≤72 h vs. >72 h); (c) sex-specific analyses.
Response: Thank you for these insightful suggestions regarding sensitivity analyses. We fully agree on the importance of addressing potential confounding factors. However, due to limitations in our data availability, detailed information on statin usage (including specific type, dosage, or duration) was not systematically collected in our cohort and could not be reliably included or adjusted for in the current analysis. We have acknowledged this limitation in the Discussion section. Regarding your other suggestions, we have performed stratified analyses based on the time from fracture to sampling (categorized as ≤72 h vs. >72 h), sex, and BMI. These results are presented in Table 4 and support the robustness of our primary findings.
Confounding & limitations
Lipid-lowering therapy and acute-phase changes after trauma are major potential confounders; integrate them explicitly in the analyses and discussion.
Response: Thank you for highlighting these important potential confounders. Regarding lipid-lowering therapy, detailed information on the use of specific medications (such as statins) was not systematically collected in our dataset and could not be directly adjusted for in the analysis. We have explicitly acknowledged this limitation in the Discussion section. Concerning acute-phase changes following fracture, we have incorporated a discussion of their potential influence on lipid measurements in the same section.
Bone mineral density (BMD) is absent but clinically important—acknowledge this limitation early, and, if feasible, include proxies (prior fracture history, FRAX items).
Response: We fully agree with your perspective, and we have acknowledged this limitation in the Discussion section.
In Abstract/Conclusions, avoid causal wording; emphasize associations and the potential influence of post-fracture physiological changes on lipid values.
Response: Thank you for your suggestion. We have modified causal wording.
Comments on the Quality of English Language
Overall, the manuscript is readable, but a light–moderate language edit would improve clarity and consistency once the structural/table issues are resolved. Key points:
Tense and voice: Use past tense for Methods/Results (“We analyzed…”, “HDL-C was…”) and present tense for established knowledge in Introduction/Discussion. Keep active voice where possible.
Response: Thank you for your suggestion. We have thoroughly reviewed the entire manuscript and made corrections accordingly.
Terminology consistency: Standardize to HDL-C/LDL-C throughout; use one term for the population (e.g., “fragility fractures” or “osteoporotic fractures”) and one for sites (“vertebral” vs “hip”) consistently.
Response: Thank you for your suggestion. We have carefully reviewed the entire manuscript and standardized the terminology throughout to ensure consistency.
Abbreviations: Define at first mention (e.g., BMD, CAD) and use consistently. Provide an abbreviation list.
Response: Thank you for your suggestion. We have defined all abbreviations at their first mention in the text and provided a list of abbreviations for reference.
Statistical phrasing: Reserve “significant” for results meeting the alpha threshold; otherwise write “higher/lower but not statistically significant (p=…)”. Prefer “associated with” over “related with.”
Response: We have revised the relevant statistical phrasing in the Results section as suggested.
Grammar & style: Check subject–verb agreement, article use (“the vertebral fracture group was…”), and prepositions (“correlation between X and Y”). Avoid unnecessary slashes (use “and/or” sparingly).
Response: We have thoroughly reviewed the manuscript, corrected grammatical and stylistic errors, and removed unnecessary slashes.
Numbers & units: Follow SI/ journal style: space between number and unit (e.g., “mmol/L”), en-dashes for ranges (e.g., 65–80), uniform decimal places, and consistent % formatting.
Response: We have thoroughly reviewed the manuscript and addressed the identified inaccuracies accordingly.
Hyphenation/line breaks: Remove PDF-induced split words (e.g., “hos-pital”), unify hyphen vs en-dash usage, and avoid mid-line hyphen breaks.
Response: Thank you for your careful review and helpful suggestions regarding formatting consistency. We have removed all PDF-induced split words (e.g., “hos-pital” → “hospital”) and standardized the use of hyphens and en-dashes throughout the manuscript. Regarding mid-line hyphenated line breaks, these were generated automatically by the journal's typesetting template. We have ensured that hyphen usage follows standard English conventions in our source document. Should the editorial team recommend further adjustments to hyphenation or line-breaking style during production, we are happy to assist accordingly.
Tables/Figures: Make captions self-explanatory and parallel in structure; ensure column headers and units are explicit and consistent.
Response: We have revised all tables
Round 2
Reviewer 2 Report
Comments and Suggestions for Authors
Thank you for your thorough and constructive revision of the manuscript. The concerns raised in the initial review regarding data consistency, clarity of methods, and appropriateness of statistical analysis have been carefully addressed. The tables are now consistent and interpretable, the methodology is clearly described with enrollment period, case definitions, and laboratory details specified, and the use of multivariable linear and logistic regression models strengthens the validity of the findings. The inclusion of stratified and sensitivity analyses further enhances confidence in the robustness of the results.
Your discussion appropriately acknowledges the limitations, including the lack of bone mineral density data and incomplete information on lipid-lowering therapy, and interprets the findings with caution. The conclusions have been reframed to emphasize associations rather than causality, which improves scientific accuracy.
The manuscript reads more smoothly after language editing, though a final round of minor editorial polishing may still be useful at the production stage to further improve clarity.
Overall, the revised manuscript is much improved and makes a valuable contribution to the literature on lipid profiles in fragility fracture patients.
Comments on the Quality of English LanguageThe manuscript is clearly understandable, and the quality of English has improved after revision. Minor editorial polishing at the production stage (e.g., consistency in tense, articles, and phrasing) may further enhance clarity, but no major language issues remain.